# Spatial Distribution of Al, Zn, Fe, As, Pb, Mn, Cr, and Cu in Surface Waters of the Urumqi River Basin, China, and Assessment of Risks to Ecosystems and Human Health

Yang Chen [1,2], Han Yang [1,2,*], Azimatjan Mamattursun [1,2], Kamila Ablikin [1,2] and Nazakat Mijit [1,2]

1   School of Geography and Science Tourism, Xinjiang Normal University, Urumqi 830054, China;
    chenyang960614@163.com (Y.C.); azimat_mamattursun@icloud.com (A.M.); kamila_518@163.com (K.A.);
    nanami_0928@163.com (N.M.)
2   Laboratory of Lake Environment and Resources in Arid Regions of Xinjiang, Urumqi 830054, China
*   Correspondence: yanghanxjnu@xjnu.edu.cn

**Abstract:** The study of pollution and risk assessment of surface water in watersheds is important for the use and management of surface water, as well as for the stability of ecosystems and human health. This study focused on a typical watershed in an arid zone, the Urumqi River basin; divided the basin into upper, middle, and lower reaches according to the main uses of the surface water in the region; and collected surface water samples from the basin. We collected 41 surface water samples from the upper, middle, and lower reaches of the Urumqi River Basin, a typical arid zone watershed. The characteristics and spatial distribution of metal elements in the surface waters of the basin were analysed, the pollution status was evaluated, and risk assessments of the effects of these metal elements on natural ecosystems and human health were carried out. The results showed that (1) the average concentration of the metal element Al in the surface water of the Urumqi River Basin was 663.73 $\mu g \cdot L^{-1}$, which was 3.3 times that of the standard limit value (200 $\mu g \cdot L^{-1}$), with an exceedance rate of 100%, and the standard deviation value was 136.05 $\mu g \cdot L^{-1}$, with a large difference in spatial distribution. Spatial distributions for Al, Cu, Cr, Fe, Mn, and Zn were higher upriver and midstream than downriver, and for Pb and As, they were higher upriver, midstream, and downriver than downriver. (2) The values of the single-factor pollution index of the metal elements Zn, As, Pb, Mn, Cr, and Cu in the upper, middle, and lower reaches of the watershed were all less than 1, which is within the safe range. The integrated pollution indexes of 0.03~0.27 were all less than 0.7, which is within the safe range, and the integrated pollution of the upper reaches was significantly greater than that of the middle and lower reaches. (3) The total ecological risk of the basin ranged from 0.09 to 13.72, which is much lower than the low-risk indicator value of 150, and the ecological risk of the upper reaches was higher than that of the middle and lower reaches. (4) The health risk assessment showed that the total health risks of the eight metal elements to adults and children showed an upstream > downstream > midstream pattern, all of which exceeded the ICRP recommended value ($10^{-5}$). The average annual total health risks of the carcinogenic metal elements chromium and arsenic to adults and children were $10^{-6}$ and $10^{-5}$ $a^{-1}$, respectively, with arsenic concentrations exceeding the ICRP recommended value ($10^{-5}$). Arsenic and lead in the surface waters of the Urumqi River Basin are important indicators of health risk and need to be prioritised as indicators for environmental risk management.

**Keywords:** surface waters; spatial distribution; Urumqi River Basin; risk assessments

## 1. Introduction

Surface waters are a key link in the terrestrial water cycle and are widely used by human populations. The water resources in arid zone watersheds are important components of the regional ecological environment and are essential to the health and livelihoods of people living in these regions [1,2]. Water shortages and pollution have become major

problems in arid zones with the acceleration of urbanisation, and the rapid expansion of industrialisation and may limit regional economic development [3]. Metal elements have a long latent period and high toxicity and are concentrated in living organisms [4]. In addition, metal elements can enter rivers and lakes via domestic sewage and industrial wastewater discharge, surface pollution from mining and agricultural activities, and atmospheric dry and wet deposition. They not only endanger the ecological balance of the watershed but can also enter the human body through the food chain, posing a serious threat to human health [5]; this is especially true for carcinogenic elements [6–9]. For example, As can damage the human nervous system and accumulate in the body, leading to chronic As poisoning. Cr can damage human kidneys and the liver, whereas Pb can affect the function of red blood cells, the brain, the kidneys, and the nervous system [6]. Metal elements also pose a significant threat to ecosystems. They are persistent in water bodies and can be amplified through food chains, affecting aquatic organisms [10]. It is therefore important to determine the mass concentrations and characteristics of metal elements in the surface waters of arid zone watersheds and to carry out risk assessments of their effects on ecosystems and human health. These risk assessments will help to clarify the current status of surface waters in arid zones during urbanisation and industrialisation and help to protect these waters, and they can be used to propose amelioration measures.

In response to this problem in China, many studies have evaluated metal element pollution in surface waters [11–13] and carried out risk assessments for both ecosystems [14–16] and human health [17–19]. Ecological risk assessments determine the potential for adverse ecological effects as a result of exposure to one or more stressors [20]. Metal elements threaten aquatic ecosystems, and their persistence in water bodies and ability to enter the food chain can be amplified, posing a potential risk to aquatic organisms [10]. Studies of metal elements in water, sediments, and soils have shown that Cd and Zn pose different levels of ecological risk [21–23]. Health risk assessments, which have been carried out since the 1980s, have linked environmental pollution to human health, quantitatively described the hazards from environmental pollution that affect the health of populations, and estimated the probability of the occurrence of hazards to humans from harmful factors. Health risk assessment is a technical basis for investigating risk from hazards and a scientific basis for prioritising hazards, exposure routes, and management, so the quantitative description of hazards to human health is of great practical importance. Health risk assessments of water bodies focus on those substances (carcinogens and non-carcinogens) present in the water that are harmful to human health [24]. Numerous studies have been carried out on the risks to aqueous environments on multiple spatial scales using the health risk evaluation model recommended by the US Environmental Protection Agency (EPA) with metal elements as water quality indicators [25]. For example, Sanjoy Shil et al. studied the water quality and pollution of the Mahananda River in West Bengal and Bangladesh. Practical experience shows that the Mahananda River is likely to be threatened by pollution, with seven sampling stations exceeding the hazard limit (HI ≤ 1) in the pre-monsoon season and two sampling stations exceeding the limit in the post-monsoon season. Ustaoğlu and coworkers investigated the status and contamination levels of 13 elements polluting seven rivers in the northeastern Giresun Basin in Turkey and showed that the levels of Al in the rivers exceeded the World Health Organization's acceptable limits and that the risk of cancer in both children and adults was lower in one river basin but higher in the remainder as a result of contamination with As, and other related studies have shown similar results [18,26–29]. Numerous studies have shown that anthropogenic activities have led to elevated concentrations of the metal elements Cr and As in soils, waters, sediments, etc., with certain potential hazards to the ecological environment and human health. For example, Li and coworkers studied the Luan River Basin and showed that heavy metal concentrations in the basin were only partially exceeded but that there was a high potential carcinogenic risk for As, which was close to the International Commission on Radiological Protection (ICRP)'s recommended levels [30–35].

In summary, surface waters worldwide are affected by metal element pollution, which not only damages the ecological environment but can also directly and indirectly endanger human health. A simultaneous ecological and health risk assessment of metal elements in surface waters will thus provide comprehensive and effective theoretical and decision bases for the risk management of surface waters. In recent years, many scholars have carried out research on the distribution characteristics, pollution evaluation, and risk assessment of metal elements in water bodies [11,32,36]. There is still a need for long-term studies on the ecological impact of metal element contamination in surface water and the potential health risks to the inhabitants of oases in watersheds with complex topography in arid zones. The Urumqi River Basin is a typical arid zone basin in western China and is the main source of water for the cities of Urumqi and Wujiaqu. These surface waters have had an important role in the development of Urumqi, the capital city of Xinjiang. With the continued expansion of the city and the rapid growth of the urban population, the already limited water resources are increasingly challenged, with decreasing per capita water holdings and pollution becoming increasingly prominent [3,37]. There is an urgent need to characterise the spatial and temporal evolution of metal elements under the influence of rapid urbanisation and to identify the ecological risks posed by metal elements and their health risks in this arid river basin from the headwaters in the upper reaches of the Urumqi River, through areas of rapid urban expansion in the middle reaches, and to areas of concentrated agricultural activity in the lower reaches. We thus need to study the spatial evolution of metal element pollution in the surface waters of the Urumqi River Basin and the response of the risks to ecological and human health as a result of urbanisation. This will help in the sustainable development of cities within this arid basin.

Ultimately, the Urumqi River Basin was selected as the study area, and the basin was divided into upper, middle, and lower reaches according to the topography and surface water use conditions of the basin. On the basis of the 41 surface water samples collected, the typical metal elements Al, Zn, Fe, As, Pb, Mn, Cr, and Cu were selected for the study, which involved (1) an analysis of the concentration characteristics and spatial distribution of metal elements; (2) an analysis of the pollution evaluation of the upper, middle, and lower reaches of the watershed under different standard values of metal elements; (3) an evaluation of the potential ecological risk of metal elements in the watershed using an ecological risk evaluation model; and (4) an analysis of the spatial distribution of the potential health risk of metal elements in the watershed using a health risk evaluation model. This study provides a theoretical basis for risk management and decision-making in the aquatic environment of the basin and provides data to support the sustainability of the aquatic ecosystems and the health of the basin's human population.

## 2. Materials and Methods

### 2.1. Overview of the Study Area

The Urumqi River Basin is located in the middle of the northern slope of the Tianshan Mountains in northwest China (Figure 1). The river crosses Urumqi County, Tianshan District, Shaibak District, and Xincheng District from south to north before entering Wujiaqu City and then flowing into Dongdaohaizi, Middong District, on the southern edge of the Junggar Basin. The basin has a typical mountain–plain–desert topography with an elevation difference of 4000 m from north to south. Soils and vegetation vary significantly with altitude, and the average annual rainfall is <200 mm [38].

We divided the basin into upper, middle, and lower reaches according to the altitude, soil, and temperature conditions and the way surface water is used. The upper reaches include the area from the source of the river to the Hongyanchi Reservoir; with Urumqi as the main city; this area is a water source. The middle reaches from the north of the Hongyanchi Reservoir to the Mengjin Reservoir include the districts of Tianshan, Shabak, Xincheng and Shuimagou, which have mixed water uses (domestic, agricultural industrial, etc.). The lower reaches, from the south of the Mengjin Reservoir to Dongdaohaizi, are mainly agricultural plains, including Middong and Wujiakou districts, and are dominated

by agricultural and domestic water use. Urbanisation in Urumqi has accelerated in recent years, with the built-up area of the city increasing from 205.01 km$^2$ in 2004 to 522 km$^2$ in 2020. The city's population has increased from 2,082,000 in the fifth census in 2000 to 4,054,400 in the seventh census in 2020. The urban area, with a population of over 3,500,000, accounts for approximately 90% of the total population. The 2021 Urumqi Statistical Bulletin recorded 125 primary schools with 254,400 students and 470 kindergartens with 114,400 children in attendance.

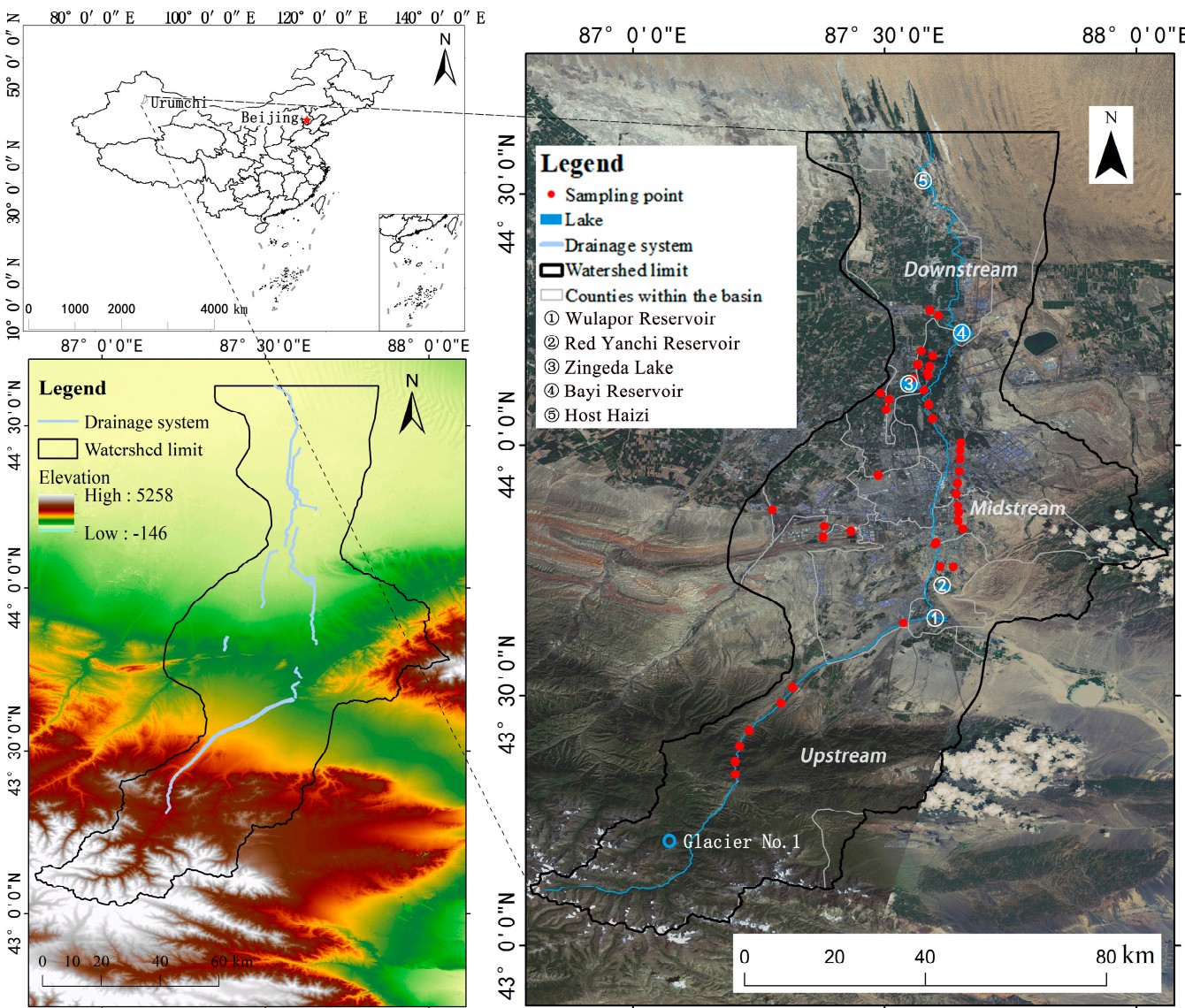

**Figure 1.** General map of the study area of the Urumqi River Basin.

## 2.2. Sample Collection, Analysis and Processing

Surface water was sampled from 41 sites along the Urumqi River from 27 to 30 October 2020. The sampling sites were selected on the basis of the distribution and accessibility of surface water in the basin. Seven sampling sites were in the upper reaches, which are mostly mountainous and have fast currents, and the number of water samples collected was limited due to the topography and other conditions; 23 were in the middle reaches; and 11 were in the lower reaches, which are mostly flat and have slow currents. At each sampling point, 500 mL of water was collected from a depth of 0.5 m below the surface of the water, with the aim of collecting water from the middle of the flowing water body.

Before sampling, the polyethylene bottles and sampling pails were washed with clean water, and the location of the sampling point was recorded using the HMS M11RTK.

*2.3. Sample Collection and Determination of Metal Element Concentrations*

The samples were brought back to the laboratory and filtered through a glass fibre membrane with a pore size of 0.45 microns. The filtrate was collected for the experiments, and a certain amount of nitric acid was added to it, with the ratio of nitric acid to filtrate maintained at 1%. The water samples were then packed into glass vials, sealed with a sealing film, and sent to the Institute of Geography in Beijing. Metal element concentrations were determined using an inductively coupled plasma emission spectrometer (Avio 500) of USA origin. The results showed that the relative standard deviation (RSD) of all the elements to be measured was less than 15%, the relative error (RE) was less than $\pm 20\%$, and the R2 of the standard curve was higher than 0.9999; thus, the data were accurate and the samples were measured.

A total of 22 metallic elements were measured, and the concentrations results of the 22 elements were analysed to select a total of 8 elements, Al, Zn, Fe, As, Pb, Mn, Cr, and Cu, for an analysis of their distribution in the surface water of the basin, their ecological impacts, and their potential risks to human health. The carcinogenic metallic elements As and Cr are the main targets of environmental risk management in surface water, and the results of a study on heavy metals in drinking water sources in Urumchi City conducted by Zurifiye Yalikun in 2021 showed that the carcinogenic metallic element As is the main indicator of health risk arising from the water quality of drinking water sources in Urumchi City [39], so this paper included the metallic elements As and Cr in the study of surface water. The concentration of the metallic element Al in the watershed obviously exceeded the standard, so analysing the spatial distribution of the metallic element Al in the watershed and evaluating its ecological and health risks are of great significance for the management of water resources in the watershed. Due to rapid industrial development in the middle part of the basin, the metal element Pb has an important industrial use in the production of rechargeable batteries, which needs to be investigated. The toxicity of the metal elements Zn, Fe, Mn, and Cu, although low, is due to the natural geology of the region.

*2.4. Assessment Methods*

2.4.1. Pollution Assessment

The upper reaches of the Urumqi River are an important source of water for domestic use in Urumqi, and the Wulapo and Hongyanchi reservoirs provide >80% of the domestic water supply. The concentrations of six metal elements (Zn, As, Pb, Mn, Cr, and Cu) in surface waters from the upper reaches were compared with the standard of the Centralised Surface Water Sources for Domestic Drinking Water (GB 5749-2006). By contrast, the surface waters in the middle and lower reaches of the Urumqi River are used for a variety of purposes, including industrial water, agricultural water, and domestic water. We thus evaluated the concentrations of the metal elements Cu, Zn, As, Cr, and Pb. The metal element Mn, which is not covered by the class III water quality standard, was adopted as the standard limiting value for additional elements in the Surface Water Quality Standard for Centralised Domestic Water Supply (GB 3838-2002).

The combined pollution index was used to evaluate metal element pollution. This index evaluates the status of metal element pollution in different river sections of the aqueous environment, emphasising the impact of the largest pollution factors on water quality through a study of all factors. The degree of pollution from all metal elements in the water body is then evaluated. The calculation is as follows:

$$P_i = \frac{C_i}{S_i} \tag{1}$$

$$P_n = \sqrt{\frac{(P_{max})^2 + (P_{ave})^2}{2}} \tag{2}$$

where $P_i$ is the single-factor pollution index of metal elements I; $P_n$ is the comprehensive pollution index of all the metal elements; $C_i$ is the measured concentration of the metal element i (mg L$^{-1}$); $S_i$ is the corresponding water quality standard (mg L$^{-1}$), here based on the GB 3838-2002 Class III Surface Water Environmental Quality Standard water quality standard as a reference; $P_{max}$ is the maximum value of the single-factor pollution index of the metal elements; and $P_{ave}$ is the average value of the single-factor pollution index of the metal elements. The pollution evaluation criteria for metal elements are shown in Table 1.

**Table 1.** Evaluation criteria for metal element pollution of water bodies [40].

| $P_i$ | $P_n$ | Pollution Level |
|-------|-------|-----------------|
| ≤1 | ≤0.7 | Safety |
| 1–2 | 0.7–1.0 | Alert |
| 2–3 | 1.0–2.0 | Light pollution |
| 3 | 2.0 | Heavy pollution |

2.4.2. Ecological Assessment Risk

The Potential Ecological Hazard Index was first proposed by the Swedish researcher Hakanson and has since been widely used to evaluate the ecological risk posed by contaminants in substrates and soils [41]. In recent years, the index has also been applied to evaluate the ecological risk to water bodies [42]. A potential risk index (RI) was calculated to analyse the potential ecological risk of six metal elements (Zn, As, Pb, Mn, Cr, and Cu) in the Urumqi River Basin. The RI was calculated by:

$$RI = \sum_{i=0}^{n} T_m^i \times \frac{C_m^i}{C_b^i} \tag{3}$$

where RI is the potential ecological risk index, $C_b^i$ is the background concentration of metal elements in the water body using the GB 3838-2002 Class III Surface Water Environmental Quality Standard water quality standard as a reference, $C_m^i$ is the metal element content of the water body at the sampling site, and $T_m^i$ is the toxicity factor for the metal elements. The toxicity factors for the heavy elements analysed here were as follows: Zn, 1; As, 10; Pb, 5; Mn, 1; Cr, 2; and Cu, 5. When RI is <150, then the ecological risk of the watershed is low; when $150 \leq RI \leq 300$, the ecological risk is medium; when $300 \leq RI \leq 600$, the ecological risk of the watershed is high; and when RI > 600, the ecological risk of the watershed is very high [43].

2.4.3. Risk Assessment for Human Health

Evaluations of the potential risk to human health from metal elements in surface waters were conducted separately for adults and children according to the US EPA methods [44]. The average daily exposure of humans to metal elements in water under different exposure pathways was first calculated for drinking water. The pathway for the measurement of dermal exposure was calculated by:

$$D_i = \frac{\rho_w \times V \times t_w \times \gamma}{m \times t_a} v \tag{4}$$

$$D_d = \frac{\rho_w \times S \times C \times t_e \times t_w \times \gamma}{m \times t_a} \times 10^{-3} \tag{5}$$

where $D_i$ is the average daily exposure dose per unit body weight of the elemental metal w via the drinking water route (mg·(kg·day)$^{-1}$), $\rho_w$ is the average mass of the elemental metal

w (mg·L$^{-1}$), $D_d$ is the average daily exposure per unit of body weight of the elemental metal w via the dermal route mg (mg·(kg·day)$^{-1}$), C is the permeation constant of water weight metal elements on the skin (cm·h$^{-1}$), and V, $t_w$, $t_a$, $t_e$, $\gamma$, m and S are the reference values in Table 2.

The potential health risks of metal elements can be classified as carcinogenic and non-carcinogenic [45]. We thus calculated the potential health risks of metal elements under different individual pathways for carcinogenicity and non-carcinogenicity.

(1)    Non-carcinogenic potential health risk calculation:

$$R_d^n = \frac{D_d}{D_d^r \times T} \times 10^{-6} \tag{6}$$

$$R_i^n = \frac{D_i}{D_i^r \times T} \tag{7}$$

where $R_d^n$ indicates the population health risk from exposure to chemical non-carcinogenic metal elements by the dermal route of penetration (a$^{-1}$), $R_i^n$ indicates the population health risk from chemical non-carcinogenic metals via the drinking water route (a$^{-1}$), $D_d^r$ is the average daily exposure reference dose of the non-carcinogenic metal w via the dermal permeation route (mg·(kg·day)$^{-1}$), and $D_i^r$ is the average daily intake reference dose of the non-carcinogenic metal w via the drinking water exposure route (mg·(kg·day)$^{-1}$]).

(2)    Carcinogenic potential health risk calculation:

$$R_d^c = \frac{D_d \times f_d}{T} \tag{8}$$

$$R_i^c = \frac{D_i \times f_i}{T} \tag{9}$$

where $R_d^c$ is the population health risk from exposure to carcinogenic metals via the dermal route (a$^{-1}$), $R_i^c$ is the population health risk from exposure to non-carcinogenic metals via the drinking water route (a$^{-1}$), $f_d$ is the average daily intake reference dose of the non-carcinogenic metal w via the dermal route ((kg·d)·mg$^{-1}$), and $f_i$ is the average daily intake reference dose of the non-carcinogenic metal w via the drinking water route ((kg·d)·mg$^{-1}$). The T reference values are detailed in Table 2, and the C, D, and f reference values for each element are shown in Table 3 [46–49].

**Table 2.** Values of parameters related to health risk assessment methods.

| Factor | Units | Adults | Children |
|---|---|---|---|
| Average daily intake via the drinking water route for humans (V) [50,51] | L·day$^{-1}$ | 2.2 | 1 |
| Duration of exposure to metal element w (tw) [52] | a | 70 | 35 |
| Exposure frequency of metal element w($\gamma$) [45] | days·a$^{-1}$ | 365 | 365 |
| Body weight per capita (m) [52] | kg | 60 | 25 |
| Average exposure time (ta) [52] | days | 12,775 for non-carcinogenic metal elements; 25,550 for carcinogenic metal elements | 12,775 for non-carcinogenic metal elements; 25,550 for carcinogenic metal elements |

Table 2. *Cont.*

| Factor | Units | Adults | Children |
|---|---|---|---|
| Area of contact between water and skin (S) [48] | $cm^2$ | 18,000 | 8000 |
| Exposure time (te) [52] | $h\cdot day^{-1}$ | 0.6333 | 0.4167 |
| Human life expectancy (T) | | 74 | 74 |

Table 3. Values of parameters related to health risk assessment methods.

| | Element | C (cm·h$^{-1}$) | D (mg·kg$^{-1}$·day$^{-1}$) | | f (kg·day·mg$^{-1}$) | |
|---|---|---|---|---|---|---|
| | | | Drinking Water Exposure | Skin Exposure | Drinking Water Exposure | Skin Exposure |
| Carcinogenic | Cr | 0.002 | 0.003 | 0.003 | 0.5 | 20 |
| | As | 0.0018 | 0.0003 | 0.000123 | 1.5 | 3.66 |
| Non-carcinogenic | Mn | 0.0001 | 0.046 | 0.0008 | | |
| | Al | 0.01 | 0.14 | 0.14 | | |
| | Cu | 0.0006 | 0.04 | 0.012 | | |
| | Fe | 0.0001 | 0.3 | 0.045 | | |
| | Pb | 0.000004 | 0.0014 | 0.00042 | | |
| | Zn | 0.0006 | 0.3 | 0.06 | | |

The reference values for C, D, and f for each element are given in Table 3 [47–49].

Related studies have shown [53] that the toxic effects of eight metal elements (Al, Zn, Fe, As, Pb, Mn, Cr, and Cu) present in water bodies on human health are additive. The overall health hazard risk evaluation model for the water environment is thus expressed as:

$$R = R_i + R_d \tag{10}$$

$$R_{Total} = \sum_{k=0}^{n} R_k \tag{11}$$

where R is the total population health risk from metals via the drinking water and dermal routes (a$^{-1}$), $R_i$ is the population health risk from metals via the drinking water route (a$^{-1}$), $R_d$ is the population health risk from metals via the dermal route (a$^{-1}$), and $R_{Total}$ is the total potential health risk from the eight metal elements (a$^{-1}$).

## 3. Results and Analysis

### 3.1. Characteristics of Metal Element Pollution in the Urumqi River Basin

Table 4 shows the maximum, minimum and average values of eight metallic elements (Al, Zn, Fe, As, Pb, Mn, Cr, and Cu) in the surface water of the Urumqi River Basin. The mean concentration of the metallic element Al in the surface water of the Urumqi River Basin was 663.73 µg·L$^{-1}$, which was 3.3 times the standard limit (200 µg L$^{-1}$), with a concentration range of 415.3–945.6 µg·L$^{-1}$, and the exceedance rate was 100%. The concentrations of the remaining seven metallic elements (Zn, Fe, As, Pb, Mn, Cr, and Cu) did not exceed their respective standard limits and were not directly contaminated, but the maximum concentration of the metallic element Pb (6.70 µg·L$^{-1}$) was close to the standard limit 1 (10 µg·L$^{-1}$) and required special attention. The standard deviations of Al, Zn, Fe, As, Pb, Mn, Cr, and Cu between different sampling points were calculated to analyse the variations of the metal elements, and the results showed that the spatial distribution of Al had the largest variation, reaching 136.05 µg·L$^{-1}$. In terms of the coefficient of variation (CoV), the CoVs of the mass concentrations of the metal elements As, Pb, Mn, Cr, and Cu in the water body were all greater than 100%, indicating that the contents of these metal elements varied greatly between different sampling points.

**Table 4.** Statistical characteristics of the concentrations of eight metal elements in the surface waters of the Urumqi River Basin ($\mu g \cdot L^{-1}$).

| | As | Al | Cr | Mn | Cu | Fe | Pb | Zn |
|---|---|---|---|---|---|---|---|---|
| Range | 0–18.9 | 415.3–945.6 | 0–1 | 0–5.9 | 0–0.9 | 5.6–55.7 | 0–6.7 | 12.3–135.5 |
| Average | 4.97 | 663.73 | 0.12 | 1.61 | 0.06 | 14.43 | 1.92 | 63.52 |
| Standard deviation | 4.71 | 136.05 | 0.3 | 1.42 | 0.18 | 8.36 | 1.91 | 24.66 |
| Coefficient of variation (%) | 94.64 | 20.5 | 255.18 | 87.86 | 303.37 | 57.94 | 99.55 | 38.82 |
| Percentage of sample points exceeding the standard (%) | 0 | 100 | 0 | 0 | 0 | 0 | 0 | 0 |
| Standard limit 1 | 50 | 200 | 50 | 100 | 1000 | 300 | 10 | 1000 |
| Standard limit 2 | 50 | - | 50 | | 1000 | - | 50 | 1000 |

Standard limit 1: China's surface water environmental quality standard (GB 3838-2002) II water standard limit value (mainly applies to the centralised surface water source for domestic drinking water in primary protection zones, rare aquatic life habitats, fish and shrimp spawning grounds, young and juvenile fish baiting grounds, etc.), including the metal elements Mn and Fe, and the standard limit of centralised surface water sources of domestic drinking water for the metal element Al in drinking water health standards (GB 5749-2022) in the concentration of standard limits. Standard limit 2: China's surface water environmental quality standard (GB 3838-2002) for Class III water (mainly applicable to secondary protected areas of centralised surface water sources for drinking water, fish and shrimp overwintering grounds, swim-throughs, aquaculture areas, and other fishery waters and swimming areas).

The spatial distribution of the concentrations of Al, Zn, Fe, As, Pb, Mn, Cr, and Cu in the surface waters of the Urumqi River Basin was obtained using GIS mapping technology (Figure 2). The analysis showed that the spatial distributions of Al, Cu, Cr, Fe, Mn, and Zn were very similar, with higher concentrations in the upstream and midstream regions and lower concentrations downstream. By contrast, Pb and As showed lower concentrations in the upstream regions but higher concentrations in the midstream and downstream regions.

The concentrations of Al in the surface waters gradually decreased with decreasing altitude, and the concentrations at all sampling points in the basin exceeded the standard. The highest concentrations of Al in the surface waters of the upper reaches of the basin may be influenced by the parent bedrock. The surface water sampling sites in the upper reaches were located on the gentle slopes in front of the mountains, where the water flows rapidly downslope over the undulating terrain, transporting minerals away from the parent bedrock. The middle reaches of the basin are mainly in the urban area of Urumqi city, and Al concentrations are also high there. The sampling sites were in the middle reaches of the Shuimo River and some urban parks. The rocks in the Urumqi River Basin are volcanic, especially those on Red Mountain, located on the eastern bank of the middle reaches of the Urumqi River. This area is a natural mountain park with high concentrations of Si and Al [54]. The significant reduction in the concentrations of Al in the downstream water bodies is most likely due to the adsorption of Al by soils and silts on the riverbed. The downstream water resources include recharge from the Erches River and water from the Urumqi River Basin, which could dilute the amount of Al in the water samples.

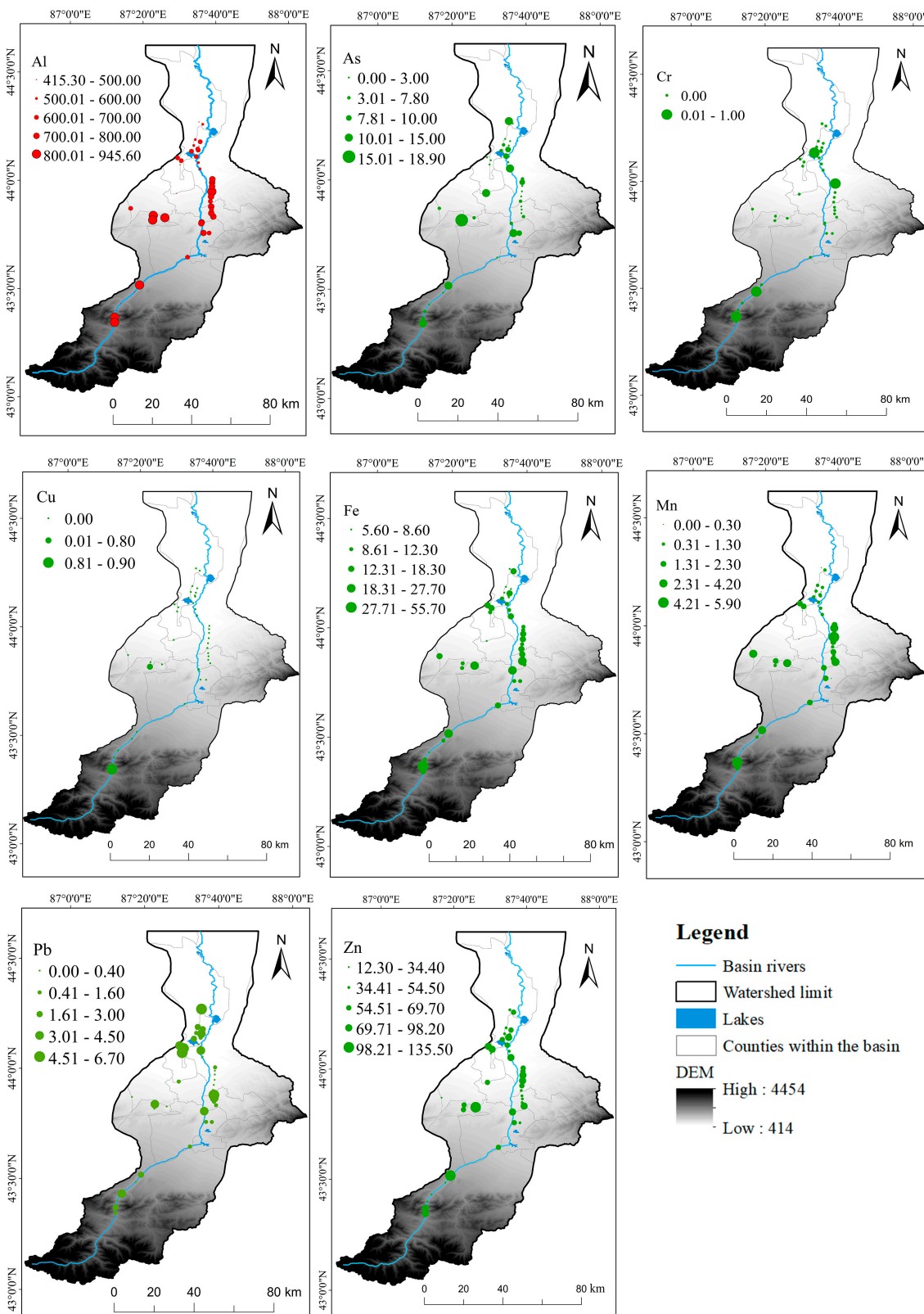

**Figure 2.** Spatial distribution of metal elements concentrations in the Urumqi River Basin. Note: Green indicates that the concentration of metal elements is less than or equal to the sanitary standard for drinking water, whereas red indicates that the concentration of metal elements is greater than the limit value of the sanitary standard for drinking water.

The spatial distribution of As concentrations in the surface water was more complex, with concentrations in the upper, middle, and lower reaches not exceeding the standard limit (>50 µg·L$^{-1}$). The highest As concentration of 13.1 µg L$^{-1}$ in the upper reaches may be related to the parent material of the watershed and the nearby car park, and it is also consistent with the results of a previous analysis of the distribution and sources of metallic elements in the surface waters of the Tianshan Mountains by Zhaoyong Zhang [55]. The highest As concentration was 18.9 µg·L$^{-1}$ in the middle reaches, and the sampling point was located near County Road 014, Salchok Village, in Toutunhe District. Its concentration may be influenced by the combination of agriculture, traffic, and industry. The downstream As concentration was up to 13.1 µg·L$^{-1}$, and the sampling site was located under the bridge of Changshanzi South Road in the Shuimo River. The area is surrounded by farmland, and the high As concentration may be related to residual fertiliser discharged into the river through ditches.

The concentrations of Pb in the surface waters were higher in the middle and lower reaches of the basin, where similar concentrations were found, but these did not exceed the standard limit. The sampling points with higher concentrations were mainly located near the Shuimo River and roads. Elemental Pb is mainly derived from the emission of Pb waste from industry and Pb-containing exhaust from motor vehicles [56,57]. We thus assume that the concentrations of Pb in surface water in the middle and lower reaches of the Urumqi River Basin may be related to industrial wastewater discharge and vehicle exhaust.

### 3.2. Ecological Risk Assessment of Metal Element Pollution in the Urumqi River Basin

3.2.1. Assessment of Metal Element Pollution

In order to assess the pollution characteristics of the sampling points in the Urumqi River Basin, we applied the Némero integrated pollution method to analyse the metal element pollution status of the sampling points in the upper, middle, and lower reaches of the basin [58]. To analyse the metal element pollutants in the waters of the study area, the single-factor pollution index method was used [58]. The results are presented in Figure 3. The single-factor pollution index values of the metallic elements Zn, As, Pb, Mn, Cr, and Cu in the upper, middle, and lower reaches of the watershed were all less than 1 and thus were within the safe range. Among them, the maximum value of the single factor index of the metal element As in the watershed was close to 04, the maximum value of the single factor index of the metal element As in the middle reaches was more than 0.4, and the maximum value of the factor index of the metal element Pb was close to 04, which requires special attention. The spatial distribution of the integrated pollution index value from ArcGIS is shown in Figure 4, and the distribution of the integrated pollution index values of the upstream, middle, and downstream reaches ranged from 0.03 to 0.27, which were less than 0.7 and within the safe range. The analysis showed that the integrated pollution of the upstream reaches was obviously larger than that of the middle and downstream reaches, and the standard limit value of the metal element Pb in the upstream reaches, a drinking water source, was also different from that of the middle and downstream reaches.

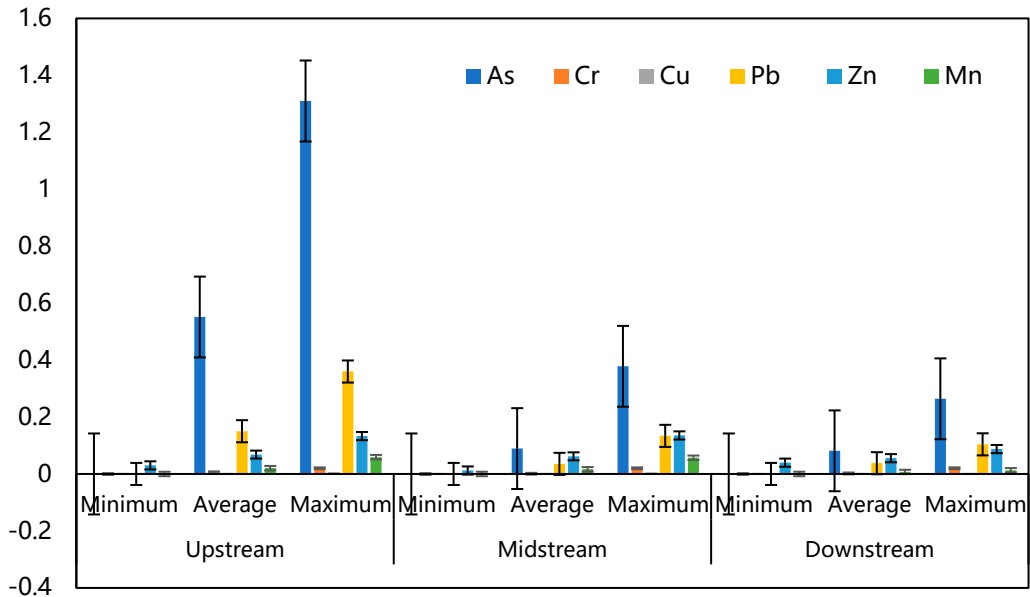

**Figure 3.** Results of the assessment of the contamination of water bodies with metal elements in different sections of the Urumqi River Basin.

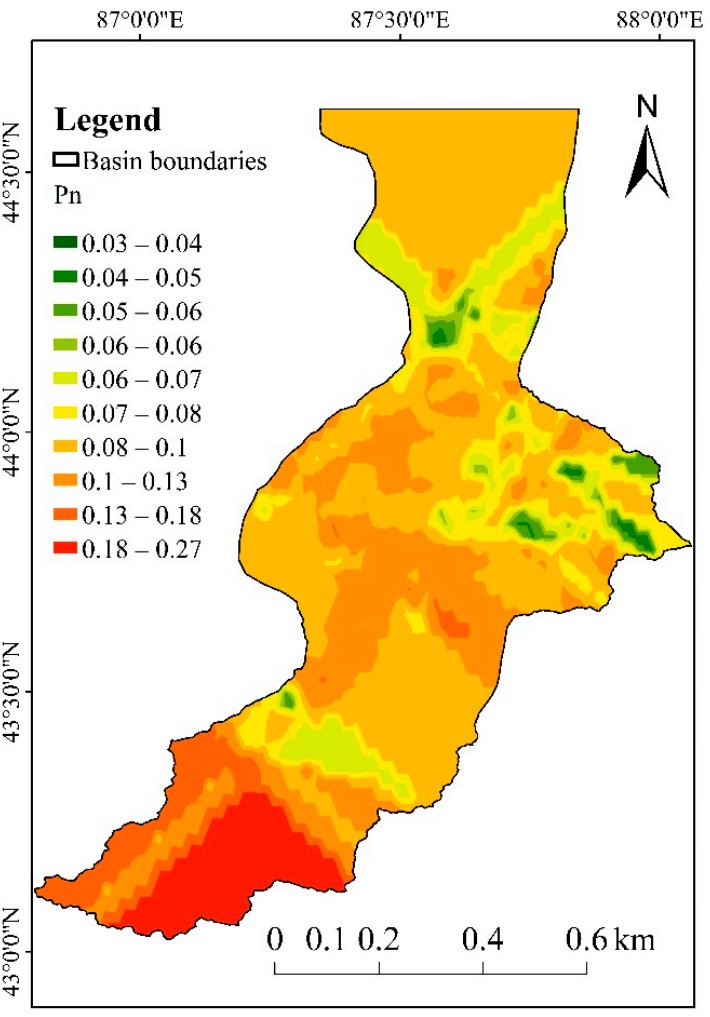

**Figure 4.** Spatial distribution of pollution assessment results in the Urumqi River Basin.

### 3.2.2. Ecological Assessment of Metal Element Pollution

We applied the potential ecological RI method to assess the risks of metal elements in the surface waters of the Urumqi River Basin at the sampling points shown in Figure 5. The total ecological risk values of As, Cr, Cu, Mn, Pb, and Zn at the sampling points ranged from 0.09 to 13.72 (all < 150), indicating that the overall ecological risk in the basin was low, although the risk was higher in the upper reaches than in the lower and middle reaches. The metal element As had the highest potential ecological risk value, followed by the metal element Pb, both of which require special attention for water catchment areas.

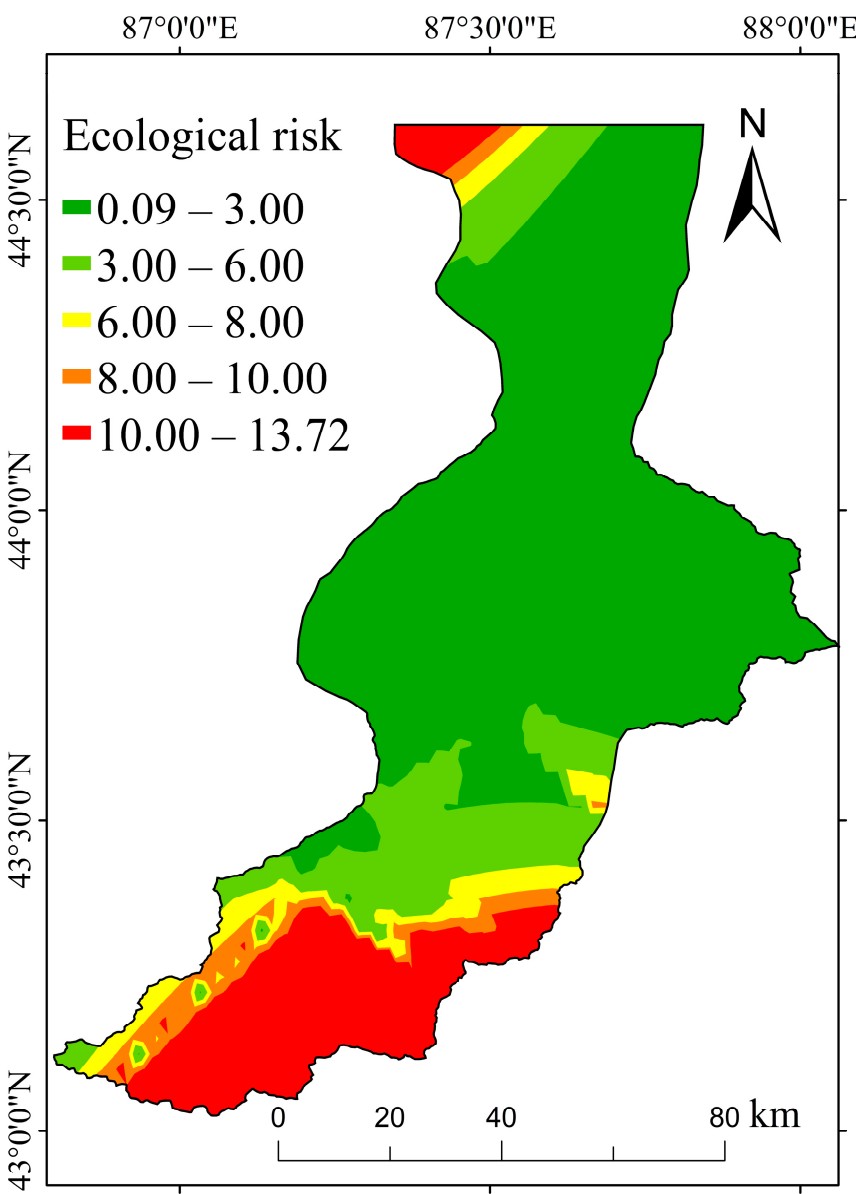

**Figure 5.** Spatial distribution of ecological risks of As, Cr, Cu, Mn, Pb, and Zn in the Urumqi River Basin.

### 3.3. *Human Health Risk Assessment of Metal Elements in the Urumqi River Basin*

### 3.3.1. Risk Analysis

We calculated the health risks and total risk (R) for Al, Zn, Fe, As, Pb, Mn, Cr, and Cu via the drinking water and dermal routes to assess the risk to human health from these metal elements in the Urumqi River Basin (Table 5). Our analyses showed that the health risks from metal elements in the river basin were higher for the carcinogenic metal elements As and Cr than for the non-carcinogenic metal elements, which exceeded or approached the ICRP and US EPA recommendations ($5 \times 10^{-5}$ and $1 \times 10^{-4}$ a$^{-1}$, respectively). The overall

risk of the carcinogenic metal element As ($8.42 \times 10^{-5}$ for adults and $6.05 \times 10^{-5}$ $a^{-1}$ for children) was higher than the ICRP recommendations. The health risk for the dermal route ($8.06 \times 10^{-5}$ for adults and $5.66 \times 10^{-5}$ $a^{-1}$ for children) was higher for adults than for children and was higher than the ICRP recommendations, and the health risk for the drinking water route ($0.35 \times 10^{-5}$ $a^{-1}$ for adults and $0.39 \times 10^{-5}$ $a^{-1}$ for children) was lower for adults than for children and was lower than the ICRP recommendations; thus, the dermal route is the main route of health risk for the metal element As. The total health risk for the carcinogenic metal element Cr for both routes ($1.01 \times 10^{-5}$ $a^{-1}$ for adults and $0.71 \times 10^{-5}$ $a^{-1}$ for children) did not exceed the ICRP recommendations, and the health risks for the drinking water and dermal routes did not exceed, but were close to, the ICRP recommendations.

**Table 5.** Assessment of the risk to human health from metal elements in the Urumqi River Basin ($a^{-1}$).

| Element | | Via Drinking Water Route | | Via Dermal Route | | Total Elemental Risk (R) | |
|---|---|---|---|---|---|---|---|
| | | Adults | Children | Adults | Children | Adults | Children |
| Carcinogenic | As ($\times 10^{-5}$) | 0.35 | 0.39 | 8.06 | 5.66 | 8.42 | 6.05 |
| | Cr ($\times 10^{-5}$) | 0.01 | 0.00 | 1.00 | 0.70 | 1.01 | 0.71 |
| Non-carcinogenic | Al ($\times 10^{-9}$) | 2.35 | 2.56 | 0.12 | 85.32 | 2.47 | 87.88 |
| | Cu ($\times 10^{-12}$) | 0.51 | 0.56 | 5.46 | 3.74 | 5.97 | 4.30 |
| | Fe ($\times 10^{-11}$) | 2.25 | 2.46 | 7.78 | 5.46 | 10.04 | 7.92 |
| | Mn ($\times 10^{-10}$) | 4.96 | 0.18 | 0.17 | 3.48 | 5.13 | 3.66 |
| | Pb ($\times 10^{-10}$) | 6.53 | 7.13 | 0.45 | 0.32 | 6.99 | 7.45 |
| | Zn ($\times 10^{-9}$) | 0.10 | 0.11 | 1.62 | 1.14 | 1.72 | 1.25 |

The risks to human health from the non-carcinogenic metal elements (Al, Zn, Fe, Pb, Mn, and Cu) were low, with the risk from total metals between $10^{-8}$ and $10^{-12}$ $a^{-1}$ (not exceeding the ICRP recommendations). The risk from total metals for children was between $10^{-8}$ and $10^{-12}$ $a^{-1}$, and the risk for adults was between $10^{-9}$ and $10^{-12}$ $a^{-1}$; the health risk for children was thus higher than that for adults. The health risk via the drinking water route was between $10^{-9}$ and $10^{-13}$ $a^{-1}$ for adults and between $10^{-8}$ and $10^{-13}$ $a^{-1}$ for children; the health risk for children was thus higher than that for adults. The health risk via the dermal route was between $10^{-9}$ and $10^{-13}$ $a^{-1}$ for adults and between $10^{-8}$ and $10^{-13}$ $a^{-1}$ for children. The health risk for adults by the dermal route was between $10^{-9}$ and $10^{-13}$ $a^{-1}$ and between $10^{-8}$ and $10^{-12}$ $a^{-1}$ for children; the health risk for children was thus higher than that for adults. The overall health risk for children from non-carcinogenic metal elements was ranked as follows: Al ($87.88 \times 10^{-9}$ $a^{-1}$) > Zn ($1.25 \times 10^{-9}$ $a^{-1}$) > Pb ($7.45 \times 10^{-10}$ $a^{-1}$) > Mn ($3.66 \times 10^{-10}$ $a^{-1}$) > Fe ($7.92 \times 10^{-11}$ $a^{-1}$) > Cu ($4.30 \times 10^{-12}$ $a^{-1}$).

We collected surface waters that were vulnerable to pollution and have a direct role in the natural environment and the livelihood of people living in the dry zone of the Urumqi River Basin. We found that the health risks posed by the carcinogenic metal elements As and Cr in the surface waters of the Urumqi Basin exceeded or approached the ICRP and US EPA recommendations. This is consistent with the results of a 2021 study of metal elements in drinking water sources in Urumqi by Yalikun [39], where the health risks were dominated by carcinogenic risks, and As was the primary risk factor. Arsenic should thus be used as a key indicator for the management of environmental risk in the surface waters of the Urumqi River Basin, and drinking water safety needs to be strengthened in the upper reaches of the river bas

### 3.3.2. Spatial Distribution of the Risks to Human Health from Metal Elements

Figure 6 clearly shows that the total risk to human health from all eight metal elements was greater for children than for adults. Spatially, the average total risk to adult health was the highest in the upstream reaches ($12.72 \times 10^{-5}$ $a^{-1}$), followed by the downstream reaches ($8.95 \times 10^{-5}$ $a^{-1}$), and it was lowest in the midstream reaches ($8.67 \times 10^{-5}$). The total risk

to children's health was also highest in the upstream reaches ($12.70 \times 10^{-5}$ a$^{-1}$), followed by the downstream reaches ($5.97 \times 10^{-5}$), and it was the lowest in the middle reaches ($5.36 \times 10^{-5}$ a$^{-1}$). The total risk to children's health exceeded the ICRP recommendation ($5 \times 10^{-5}$ a$^{-1}$). The maximum values of the total health risk from metal elements for adults showed a pattern of midstream > upstream > downstream, and those for children showed a pattern of upstream > midstream > downstream; the total health risk exceeded the ICRP recommendations.

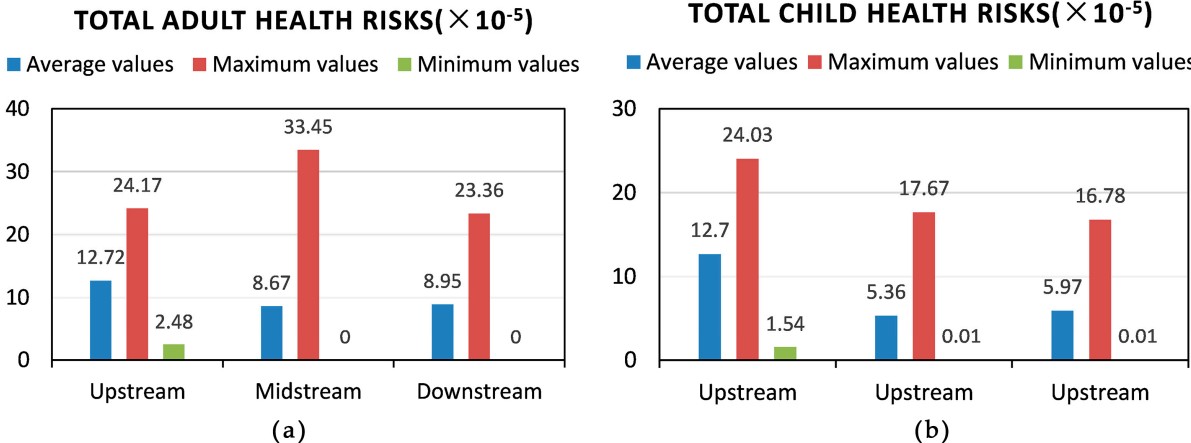

**Figure 6.** Total health risks to adults (**a**) and children (**b**) from metallic elements in the upper, middle and lower Urumqi River basin (a$^{-1}$).

These assessments of the ecological and health risks from metal elements in the Urumqi River Basin provide important data to support our understanding of the characteristic patterns of the spatial and temporal evolution of metal element pollution, the status of metal element pollution, the ecological risk, and the risk to human health in the basin. Our study covered the full extent of the river basin, from the source area in the upper reaches of the Urumqi River to the area of rapid urban expansion in the middle reaches and the area of concentrated agricultural activities in the lower reaches, which is under the influence of rapid urbanisation.

### 3.3.3. Spatial Distribution of the Risks to Human Health from As

We used As as a priority indicator in a detailed analysis of the spatial distribution of the health risk of metal elements for children and adults because it was the main element posing a risk to health for both age groups (Figure 7). The risk to human health from As was the highest in the Urumqi River Basin, and As was the primary factor generating a health risk. Spatially, health risks were present in the upper, middle, and lower reaches of the basin, with a dispersed distribution, and they exceeded the ICRP recommendations. The health risks to children from As exceeded the ICRP recommendations at 21 sampling sites: four upstream sites (57.14% of the total upstream sites), twelve midstream sites (52.17% of the total midstream sites), and five downstream sites (45.46% of the total downstream sites). The health risks to adults from As exceeded the ICRP recommendations at 18 sampling sites: three upstream sites (42.86% of the total upstream sites), eleven midstream sites (47.82% of the total midstream sites), and four downstream sites (36.36% of the total downstream sites). Exceedances of the metallic element As in the middle and lower reaches of the basin are particularly serious, and most of the sampling points in the middle and lower reaches of the basin are located in areas of industrial and agricultural development, so their concentrations may be influenced by the anthropogenic environment. It has been shown that arsenic and its soluble compounds are toxic. With the development of industries, such as metallurgy and the chemical industry, and the development of poor mines, arsenic, accompanied by the main elements, can enter the wastewater, and the arsenic level is quite high. In addition, the excessive use of agricultural fertilisers has an impact on the mass

concentration of the metal elements Cu and As; in order to achieve high quality and a high yield in agriculture, excessive pesticides and fertilisers are used, and many compounds containing the metal element arsenic are added to pesticides, such as herbicides and other pesticides, which enter the water body through the ground surface and underground runoff and other routes to cause the pollution of surface water [59].

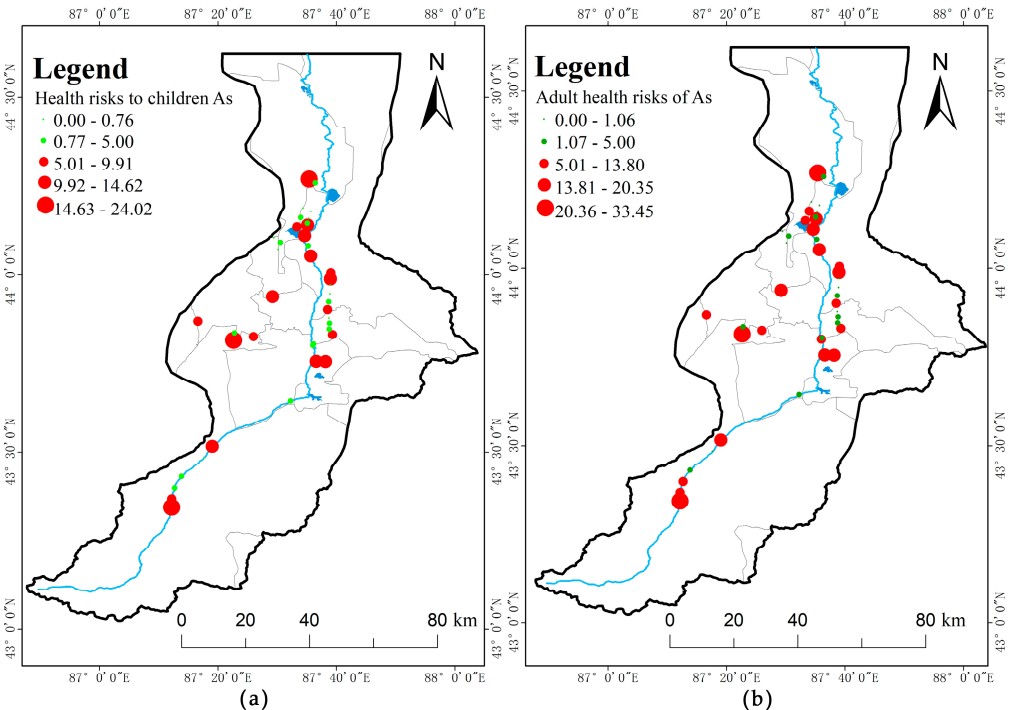

**Figure 7.** Spatial distribution of child (**a**) and adult (**b**) health risks from metal element As in the Urumqi River Basin.

The following recommendations are made to strengthen the prevention and control of metal element pollution and health risks in the water resources of the Urumqi River Basin: (1) conduct medium-and large-scale research to accurately analyse the sources of metal element pollution; (2) take effective measures to curb sources of metal element pollution and reduce biotoxicity; and (3) adopt prevention and control and remediation measures for water resources in zones at risk of metal element pollution.

## 4. Conclusions

The average concentration of the metal element Al in the surface water of the Urumqi River Basin was 663.73 μg·L$^{-1}$, which was 3.3 times the standard limit (200 μg·L$^{-1}$), and the concentration ranged from 415.3 to 945.6 μg·L$^{-1}$, with an exceedance rate of 100%, and the concentration of the remaining seven metal elements (Zn, Fe, As, Pb, Mn, Cr, and Cu) did not exceed their respective standard limits. The results of the standard deviations showed that the spatial distribution of the metallic element Al was the most variable, reaching 136.05 μg·L$^{-1}$. The coefficients of variation of the mass concentrations of the metallic elements As, Pb, Mn, Cr, and Cu in the water body were all greater than 100%, indicating that the contents of these metallic elements varied greatly between different sampling points. The spatial distributions of Al, Cu, Cr, Fe, Mn, and Zn were very similar, with higher concentrations in the upstream and midstream areas and lower concentrations downstream. The spatial distributions of Pb and As were more complex, with lower concentrations in the upstream area and higher concentrations in the middle and downstream areas.

The values of the single-factor pollution index of the metallic elements Zn, As, Pb, Mn, Cr, and Cu in the upper, middle, and lower reaches of the basin were all less than 1, which is within the safe range. The maximum value of the single factor index of the metal element As was close to 04, the maximum value of the single factor index of the metal element As

in the middle reaches was more than 0.4, and the maximum value of the factor index of the metal element Pb was close to 0.4, requiring special attention. The combined pollution index values of 0.03 to 0.27 were less than 0.7, which is within the safe range, and the combined pollution in the upper reaches was significantly greater than that in the middle and lower reaches. The overall ecological risk ranged from 0.09 to 13.72. This is well below the value of 150, which indicates low risk. Therefore, although it was higher in the upper reaches than in the middle and lower reaches, the ecological risk in the catchment was low.

The health risk assessment showed that the dermal route of exposure was the main pathway for the annual average health risk from the metal elements, with the total annual average health risk for adults and children on the order of $10^{-6}$ to $10^{-5}$ $a^{-1}$ for the carcinogenic metals Cr and As and on the order of $10^{-13}$ to $10^{-9}$ $a^{-1}$ for the non-carcinogenic metals Al, Zn, Fe, Pb, Mn, and Cu. Therefore, carcinogenic risk is the main factor affecting the average annual health risk for both adults and children. The total health risk from metal elements for adults was highest in the upstream reaches, followed by the downstream reaches, and it was the lowest in the midstream reaches, while the total health risk for children was highest in the upstream reaches, followed by the midstream reaches, and it was the lowest in the downstream reaches, with all upstream, midstream, and downstream total risks exceeding the ICRP recommendations. As is the primary indicator of health risk in the surface waters of the Urumqi River Basin and can thus be used as a priority indicator for environmental risk management. The concentrations of As in the surface waters of the Urumqi River Basin thus pose significant health risks and require special attention.

**Author Contributions:** Conceptualization, H.Y. and Y.C.; methodology, Y.C.; software, Y.C.; validation, Y.C., H.Y. and A.M.; formal analysis, Y.C.; investigation, Y.C.; resources, Y.C.; data curation, Y.C.; writing—original draft preparation, Y.C.; writing—review and editing, Y.C.; visualization, Y.C.; supervision, H.Y. and K.A.; project administration, H.Y. and N.M.; funding acquisition, H.Y. All authors have read and agreed to the published version of the manuscript.

**Funding:** Grant-in-Aid: This research was funded by the National Natural Science Foundation of China (42061007, 42261058).

**Data Availability Statement:** Not applicable.

**Conflicts of Interest:** The authors declare no conflict of interest.

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
