# Peer review of "Spatial Distribution of Al, Zn, Fe, As, Pb, Mn, Cr, and Cu in Surface Waters of the Urumqi River Basin, China, and Assessment of Risks to Ecosystems and Human Health"

_water, doi:10.3390/w15173079_

Round 1

Reviewer 1 Report

The manuscript describes the spatial distribution of heavy metals in surface waters of the Urumqi River basin and the assessment of risks to ecosystems and human health.
The manuscript consists of abstract/keywords, introduction, 1. Overview of the study area, 2. Sample collection, analysis and processing; 2.1. Collection of samples and determination of concentrations of heavy metals, 2.2. Assessment methods
2.2.1. Pollution assessment, 2.2.2. Ecological assessment risk, 2.2.3. Human health risk assessment, 3. Analysis and discussion, 3.1. Characteristics of heavy metal pollution in the Urumqi River Basin, 3.2. Ecological assessment of heavy metal pollution in the Urumqi River Basin, 3.2.1. Assessment of heavy metal pollution, 3.2.2. Ecological assessment of heavy metal pollution, 3.3. Human health risk assessment of heavy metals in the Urumqi River Basin, 3.3.1. Risk analysis and discussion, 3.3.2. Spatial distribution of risks to human health from heavy metals, 3.3.3. Spatial distribution of risks to human health from As , 4. Conclusions and references.
The following adaptations are suggested for manuscript authors:

1 – It is suggested to adapt the structure of the manuscript to:
Abstract keywords
1. Introduction
2. Materials and methods
2.1 Overview of the study area
2. 2 Sample collection, analysis and processing
2.3. Sample collection and determination of heavy metal concentrations
2.4. Assessment methods
2.4.1. Pollution assessment
2.4.2. Ecological assessment risk
2.4.3. Risk assessment for human health,
3 Results and discussion
3.1. Characteristics of heavy metal pollution in the Urumqi River Basin
3.2. Ecological assessment of heavy metal pollution in the Urumqi River basin
3.2.1. Assessment of heavy metal pollution
3.2.2. Ecological assessment of heavy metal pollution
3.3. Human health risk assessment of heavy metals in the Urumqi River Basin
3.3.1. Risk analysis and discussion
3.3.2. Spatial distribution of risks to human health from heavy metals
3.3.3. Spatial distribution of risks to human health from As
4. Conclusions
References.
2 – The expression heavy metals is no longer used, if possible replace it with toxic metals. It is observed that the scientific community adopted the ??? = metal toxicity factor, reinforcing that the ideal is to talk about the toxicity of a metal.
3 – In which part of the world is the Urumqi River Basin located?
4- While reading the manuscript, it is observed that the authors studied the presence of the following toxic metals: Al, Zn, Fe, As, Pb, Mn, Cr and Cu. Therefore, it is suggested to authors that these metals be placed in the text, for example, instead of “Spatial distribution of heavy metals in surface waters of the Urumqi River basin and assessment of risks to ecosystems and human health” the title could be : Spatial distribution of Al, Zn, Fe, As, Pb, Mn, Cr and Cu in surface waters of the Urumqi River Basin - China and assessment of risks to ecosystems and human health. It is observed that the authors did not carry out the study for all toxic metals, such as mercury, cadmium, highly toxic metals.
5 – The authors assume that the metals studied are toxic. However, it is suggested that a discussion of the toxicity of these metals be included in the manuscript. For example, iron is also essential in the metabolism of humans. Zinc also participates as a cofactor in human metabolism. Therefore, it is essential to define when they will be considered toxic.
6 – It is suggested that the manuscript undergo a review to make the text more friendly to those interested in the topic. A sample of this need is the excerpt (lines 103 to 105): “The important role of surface waters in the basin for the hydrographic basin and the complexity of using surface water resources in the basin make the study of heavy metals in surface waters of the basin particularly important. ” The word watershed appears four times.
7 – The study area is part of materials and methods, it is suggested that item 1 – Overview of the study area be integrated into it.
8 - Questions about sample collection:
8.1 - Were the collected samples acidified?
8.2 - Why were the samples transferred to a glass vial? It is known that the pores contained in the surfaces of flasks, plastic or glass, absorb part of the metallic ions contained in water samples.
8.3 – How long did it take between sample collection and sample analysis?
8.4 – The availability of GPS coordinates of the sampling points is very important.
9 – What is the country of origin of a DIONEX inductively coupled plasma mass spectrometer Model ICS-900?
10 – It is imperative that the authors provide parameters for validation of the analysis methodology, such as: Specificity/Selectivity, Response Function (analytical curve), Linearity, Sensitivity, Accuracy, Precision (repeatability, intermediate precision and reproductility), Limit of Detection (LD ), Limit of Quantification (LQ) and Robustness.

11 – What criteria were used to choose the eight metals that were selected to analyze the distribution of heavy metals in the surface waters of the basin, their ecological impact and potential risks to human health?

12 – In lines 354 to 355: “This area is surrounded by agricultural land and the high concentrations of As may be related to the release of residual chemical fertilizers into the river through ditches”. Is the high concentration of As related to the use of chemical fertilizers or the use of pesticides?

13 – What would explain the increasing concentrations of arsenic along the river? Is the explanation just the use of agrodefensive?

14 – In figure 3, it is suggested to adapt the colors of the subtitles. They are not different enough to distinguish metal concentrations.

Yours sincerely,

Author Response

Dear reviewers
Hello, thank you very much for your comments on my paper, I have revised it, please check, thank you!
My sincerity

Reviewer 2 Report

In the current study, totally 41 surface water samples were collected from the upper (7), middle (23) and lower (11) reaches of the Urumqi River Basin. Accordingly, the concentration of eight metal elements (Al, Zn, Fe, As, Pb, Mn, Cr, Cu) of the surface water were analyzed. Based on the above metal contents, three evaluation methods (pollution index, ecological risk assessment and health risk assessment) were used for evaluating the pollution and risks status. It also led to a series of conclusions. However, these results are not reliable.

The MS was not written in a rigorous manner and needs to be reorganized scientifically according to journal guidelines. The language and format of the MS need to be carefully revised before the paper submission. Some examples but not limited to these are shown below.

1.      L1, L22-23, L184, ect. The author may misunderstand the heavy metals. Is aluminum (Al) one of heavy metal elements?

2.     Most of references were listed and cited in an incorrect format. Also, the content does not agree with the cited references (L69-74, L88-92, L93-98). The author needs to deal with this problem seriously.

3.     A large number of sentences are incomplete. L55-56, L67, L123-127, L130, L133……

4.     In Abstract and other sections, such as, L22-23, L517-518. “The average concentrations of eight heavy metals in the surface waters of the Basin were in the order Al > Zn > Fe > As > Pb > Mn > Cr > Cu, with the concentrations of Al and As exceeding the standard limits1.” There is nonsense and inappropriate to compare different metal elements. Because the concentration limit values in a same standard differ by several orders of magnitude, such as Zn and As, Pb and Cu.

5.     In the whole manuscript, “standard limits1 (GB 5749-2006)” was used for evaluating the upper reaches of the Urumqi River. However, the application of this standard in this study is inappropriate. Therefore, the corresponding conclusions are also wrong (Abstract L21-31; section 2.2.1; section 3.1, 3.2.1; section 4).

The reasons are below: (1) This standard (GB 5749-2006) is used for drinking water quality, not for the surface water source of drinking water. (2) In this standard, it is also stated that when surface water is used as the source of drinking water, it should meet the requirements of GB 3838. (3) GB 5749-2006 has been replaced by the new standard GB5749-2022.

6.     Also, the application of GB 3838-2002 in this study is confused. Therefore, the ecological risk assessment results may be unreliable (L30-33). The environmental quality standards for surface water is divided into five classes (I, II, III, IV, V), and the standard value of the corresponding class is implemented for different functional classes. The authors should choose the corresponding standard reference values according to the environmental functions and conservation objectives of the Urumqi River Basin, rather than mixing the standard reference values. L223, Local background concentration of heavy metals should be considered, besides for standard values.

7.     In section 2.1, there are no details of measurement methods. The method involved in this paper does not use the national standard method corresponding to the evaluation standard. How to ensure the reliability of the measurement results? All instruments used in experiment should contain a description of model, manufacturer and country. Is the concentration of Cr totally content, or just content of Cr6+? The details of heavy metal content analysis must be clarified.

8.     L184-185 “A total of 22 heavy metals were determined and eight were then selected to analyze the distribution of heavy metals in the surface waters of the basin,” Why choose these eight metals, not Cd, Hg and others?

9.     In section 3.2-3.3, there is no discussion. The section 3.2.3 was supported by only one short paragraph. Completely improved discussion must be needed. The author does not seem to be clear about the difference between the results and the discussion.

10.  In section 3, the serial numbers of tables and figures in the text are confused. There is no Figure 3, but Figure 5 with two. L417(Table 6 ? ); L467 (Table 7 ?) Some figures were not shown in the text.

“Mid-stream” on right of Figure 7 should be “Upstream”.

What is “Multiples” or “Multiplier” in table 4? Standard deviation or variance should be included.

11.  In the whole MS, the author repeatedly emphasized the difference of water use in the upper, middle and lower reaches. For example, L152-153, “We have divided the basin into upper, middle and lower reaches based on altitude and soil and temperature conditions and the way surface water is used.”; L481-485, “Our study covered the full extent of the river basin, from the source area in the upper reaches of the Urumqi River to the area of rapid urban expansion in the middle reaches, and then to the area of concentrated agricultural activities in the lower reaches under the influence of rapid urbanization.”

Therefore, why is there no difference in the relevant parameters of the whole river basin (upper, middle and lower reaches)? (Table 2 and 3)

12.  And whether the parameter values in the tables 2 and 3 are reliable at present? All the cited reliable references needed to be shown in tables.

tw is 70 for adults, 35 for children, why?

ta = tw/365 ? If so, the formulas (4) and (5) should be replaced.

D and F values for carcinogenic, and C value for Pb in Table 3 should be checked for correctness.

Abbreviations and units should be consistent throughout the text (F, fi, fd and their units).

Author Response

(The authors gave the same response as above.)

Reviewer 3 Report

The article submitted for review concerns heavy metals toxic to human health found in the surface waters of the Urumqi river basin (China). The manuscript contains very interesting and valuable research results, but it was prepared very poorly. Papers should be thoroughly edited before publication. Below are some notes to help you do that.

General notes

Numbering of chapters and subchapters - needs improvement. Titles of chapters and subchapters - must be clearly and precisely formulated, they should refer to the content contained therein.

Abstract

It is too long - it should be shortened, and only the most important information and conclusions should be given.

Introduction

Too long, no clear and precise justification for the purpose of the research. Lack of a clearly formulated goal of work. It contains information about the research area that should be included in the "Study area" chapter.

Material and methods

No clearly separated chapters Material and methods and Study area (may be included in the chapter Material and methods).

Why is there such a large disproportion between the number of samples taken: 7 and 23? Differences in the number of trials should be better explained.

Results and discussion

This is more of an overview of the results than a discussion. No comparison with results from other regions of China or other parts of Asia or the world.

References

Prepared very carelessly, it should be adapted to the standards of the journal.

Figures and tables

Figures and tables – incorrectly quoted, e.g. Table 5, Figure 4, or no citation, e.g. Figure 5.

No table 6 (quoted in 3.3.1. Risk…. - line 417).

Figure captions - imprecise, incorrectly worded, e.g. Figure 3.  

Figure 1 - the markings are hardly visible, especially in the figure on the right

Figure 2 - geographic coordinates poorly visible

Figure 6 - should be omitted as it shows the same results as Table 5.

Figure 7 - errors in explanations.  

Other notes:

Line 94

 "Li and coworkers studied..." - no citation, what article is it about?

Line 227-228

Why are the results included here (chapter) if this is a description of the methods?

Line 378-379

“The results are shown in Table 5.” – Table 5 shows something completely different!

Line 467

Table 7 or Figure 7 ???

Line 491 and 513

Figure 6 and Figure 5 - ????

Line 505-511

These recommendations should be included in the conclusions section.

Author Response

(The authors gave the same response as above.)

Reviewer 4 Report

This manuscript focuses on a typical watershed in the arid zone, the Urumqi River basin, and divides the basin into upper, middle and lower reaches according to the main uses of surface water in the region, and collects surface water samples from the basin. Authors collected surface water samples from the upper, middle and  lower reaches of the Urumqi River Basin, a typical arid zone watershed. The characteristics and 1spatial distribution of heavy metals in the surface waters of the basin were analysed,, evaluated the pollution status and carried out risk assessments for the effects of these heavy metals on natural ecosystems and human health.

I consider that this is a study of great interest well designed and that it provides important and interesting results and conclusions from the scientific point of view. However, the work requires some improvements before being accepted for publication:

1) In the introductory section the authors must indicate the main novelties that their study brings.

2) The Section 2.1. "Sample collection and determination of heavy metal concentrations", must be improved intensively. Please, indicate the sample preparation protocols, dilutions, patterns used, detection limits for each element analyzed, etc.

I recommend that this manuscript be reviewed by a native English colleague to avoid some spelling and grammatical errors.

Author Response

(The authors gave the same response as above.)

Round 2

Reviewer 1 Report

Dear editor,
Although the authors have responded to some of the suggestions/doubts, the manuscript is not yet ready for publication. Attached is a file with the observations.
Yours sincerely,

Author Response

Dear reviewer
Hello, thank you very much for your comments on my paper, I have revised it, please check it, thank you!
Yours sincerely.

Yang Chen

Reviewer 2 Report

The manuscript could be further improved by following aspects

1.     Some of the content is seriously inconsistent with the cited references. The author needs to deal with this problem seriously. This comment was noted in the first review, hereno longeretc.

2.    Author gave an answer in the cover letter: “The measured Cr is the total content, excluding Cr6+.”It is very confusing. Additionally, the Cr6+ content in national standard (GB3838-2002) was used to assess the total Cr content in this current MS, it is obviously inappropriate.

3.    It is still not appropriate to apply this standard (GB5749-2022) in this study.

4.    “A total of 22 heavy metals were determined and eight were then selected to analyze the distribution of heavy metals in the surface waters of the basin,” Why choose these eight metals, not Cd, Hg and others?

5.    In section “2.3 Determination of heavy metal concentrations”, the following details were added by authors in the revised MS. However, this information (F-,Cl-,NO3-,SO42-K+,Ca2+,Na+,Mg2+) is not shown anywhere in the results of MS. Information about heavy metals is not reflected. 

The water samples were then packed into glass vials, sealed with a sealing film and sent to the Institute of Geography in Beijing. The anion concentration (F-,Cl-,NO3-,SO42-) was determined using an ion chromatograph (Thermo Fisher, DIONEX ICS-900) of U.S.A. origin, and the cation concentration (K+,Ca2+,Na+,Mg2+) and metal element concentration were determined using an inductively coupled plasma emission spectrometer (Avio 500) of U.S.A. origin.”

6.    “3. Results and analyses” should be “Results and discussion”. Or “4. Discussion” should be added after the section 3. 

7.    The section 3.2.3 was supported by only one short paragraph. Completely improved discussion must be needed.

8.    According to the different uses of surface water in the region, why is there no difference in the values of relevant parameters on the whole river basin (upper, middle and lower reaches)? (Table 2 and 3)If such a difference exists, it would be a highlight of this MS.So, what causes the parameter values to be different?

9.    Therefore, the scope of application of this method (human health risk assessment in this MS) and its advantages and disadvantages should be provided in Section Introduction.

The value ranges and limits of all parameters in table 2 and 3 should be showed in Section Methods based on a series of authoritative journals and materials.

10. tw is 70 for adults, 35 for children. why not 7 or 18 or another value for children?

11. Table 2, ta is the same for adults and children, why?

“ta(days) 12775 for non-carcinogenic metal elements; 25550 for carcinogenic metal elements.”Is this equivalent to 35 yearsfor non-carcinogenic metal elements and70 years for carcinogenic metal elements?

12. Abbreviations and units should be consistent throughout the text (F, fi, fd and their units).F (kg·day−1·mg −1) in table 3,fd,fi(kg·d)·mg−1 in text, so, which is the right unit ?

Author Response

(The authors gave the same response as above.)

Reviewer 3 Report

The submitted manuscript has not been corrected well, there are still serious errors that make the article in its current form unpublished. For example - in the responses, the authors state that part 2 is "2.Material and methods", and in the manuscript there is: "2. Sample collection, analysis and processing”.

 In addition:

1) there are still two Figures No. 5;

2) Figure 1 - not well corrected, e.g. numbers 1, 2, 3, 4 are not marked, geographical coordinates are still illegible next to the figure in the upper left corner,

3) there are misspellings in the figures, e.g. Figure 6: "elements" and "element"; Figure 7: "Mid-stream" and "Midstream" - unify the spelling of words.

These are just examples of items that have not been corrected.

Please re-read the comments that were submitted earlier and correct the article according to these comments.

Author Response

(The authors gave the same response as above.)

Round 3

Reviewer 3 Report

Not all figures have been quoted in the text - figures 1 and 6, and corrected - figure 6.